# KStable: A Computational Method for Predicting Protein Thermal Stability Changes by K-Star with Regular-mRMR Feature Selection

**DOI:** 10.3390/e20120988

**Published:** 2018-12-19

**Authors:** Chi-Wei Chen, Kai-Po Chang, Cheng-Wei Ho, Hsung-Pin Chang, Yen-Wei Chu

**Affiliations:** 1Department of Computer Science and Engineering, National Chung Hsing University, Kuo Kuang Rd., Taichung 402, Taiwan; 2Institute of Genomics and Bioinformatics, National Chung Hsing University, Kuo Kuang Rd., Taichung 402, Taiwan; 3Ph.D. Program in Medical Biotechnology, National Chung Hsing University, Kuo Kuang Rd., Taichung 402, Taiwan; 4China Medical University Hospital, No. 2, Yude Rd., Taichung 404, Taiwan; 5Biotechnology Center, Agricultural Biotechnology Center, Institute of Molecular Biology, Rong Hsing Research Center for Translational Medicine, National Chung Hsing University, Kuo Kuang Rd., Taichung 402, Taiwan

**Keywords:** protein thermostability, single-site mutations, machine learning, feature selection, hill-climbing algorithm

## Abstract

Thermostability is a protein property that impacts many types of studies, including protein activity enhancement, protein structure determination, and drug development. However, most computational tools designed to predict protein thermostability require tertiary structure data as input. The few tools that are dependent only on the primary structure of a protein to predict its thermostability have one or more of the following problems: a slow execution speed, an inability to make large-scale mutation predictions, and the absence of temperature and pH as input parameters. Therefore, we developed a computational tool, named KStable, that is sequence-based, computationally rapid, and includes temperature and pH values to predict changes in the thermostability of a protein upon the introduction of a mutation at a single site. KStable was trained using basis features and minimal redundancy–maximal relevance (mRMR) features, and 58 classifiers were subsequently tested. To find the representative features, a regular-mRMR method was developed. When KStable was evaluated with an independent test set, it achieved an accuracy of 0.708.

## 1. Introduction

A mutation in a protein may alter its function and/or stability via a change in conformation. Because stability is an important characteristic of proteins that impacts, for example, drug development, protein structure determination, and enzyme mechanics and dynamics, knowledge of a protein’s stability is often required for industrial and research applications. Previously, to acquire information on the stability of a protein, complicated and expensive experiments were performed. Recently, however, bioinformatics, biomathematics, and biostatistics have been used to predict mutagenesis-induced changes in protein stability [1,2]. The predictive accuracy of such computational tools has benefited from recent experimental thermodynamic studies, which have provided a large dataset, including structural and functional information, important for stability prediction [3].

The currently available computational tools used to predict changes in protein stability are of three types. First are the tools that use protein structures. These tools may incorporate various energy functions, and generally are very accurate owing to the abundant amount of information that the characterized structure surrounding the mutation site provides [4,5,6,7,8,9,10,11]. Second are tools that use the protein sequences and the physicochemical properties of the residues. Because the tertiary structures of most proteins have not been elucidated, tools that rely exclusively on sequence data can be used in more cases than can those that rely only on structural data. However, because sequences by themselves do not provide structural data, the inherent accuracy of such tools is generally less than that required for stability prediction [12,13,14,15,16,17]. Third are the tools that integrate a variety of predictive methods. These tools use machine-learning algorithms to produce a single, accurate prediction based on multiple output data [18,19,20,21].

Because it has been difficult to improve the accuracy of algorithms using only sequence information for prediction, such programs might be improved with the use of other available information concerning the stability of a targeted protein. Addition of thermostability information should also improve the accuracy of integrated predictive tools that use sequence-based information. Development of new computational methods that allow for prediction of changes in protein thermostability should involve new input features and learning algorithms that are not currently used. Structure-based prediction tools usually use previously developed energy-based functions as input and support-vector machines or neural networks as learning algorithms. In the absence of structural information, sequence-based prediction tools use the amino-acid composition, physicochemical properties of the residues, and evolution information derived from the protein sequence as input and a support-vector machine as the learning algorithm. The tool MuStab was the first to include feature selection as a third computational element [22,23].

Herein, we describe a new computational tool, named KStable, to predict changes in protein thermostability upon mutation using sequence-based information. This method was trained using two types of input data: basis features and minimal redundancy–maximal relevance (mRMR) features [24]. The basis features include the location of the mutation, the wild-type and replacement amino acids, and experimental information, including temperature and pH. The mRMR features are the various physicochemical characteristics selected by the mRMR-feature selection method [24,25], which is based on the filter approach. Compared with the wrapper approach, feature selection methods based on a filter approach usually cannot identify and eliminate redundant features, whereas mRMR can select feature sets with maximum correlations and minimum redundancies. Three types of features, namely 2-mRMR, diff-mRMR, and all-mRMR, were included in the development of KStable. To enhance and validate the effectiveness of the mRMR-feature selection method, we developed the regular-mRMR method and used the hill-climbing algorithm [26] to produce the best combination of features. For development of the learning algorithm, 58 machine-learning classifiers in seven categories (bayes, function, lazy, meta, misc, rules, and trees) in Weka [27,28] were tested for model selection. The version of KStar [29] that used the lazy category was chosen because it had the best performance.

The performance of KStable is better than that of any other tool used to predict changes in thermostability based on sequence information, with the exception of EASE-MM [13]. KStable also performs better than certain structure-based tools. KStable has an important advantage over EASE-MM in that it incorporates temperature and pH values as inputs. Moreover, the average prediction time for each inputted mutation is 10 min for EASE-MM but < 1 min for KStable. To compare other feature-selection methods that use the filter approach, we also assessed the effects of the 30 most important features during the training of KStable. KStable is available at http://predictor.nchu.edu.tw/KStable.

## 2. Materials and Methods

The workflow that developed KStable is diagramed in Figure 1. After collecting and processing data from Prothem, we assessed which basis and mRMR features and which of 58 machine-learning classifiers should be incorporated to maximize the accuracy of KStable.

### 2.1. Dataset

The training set that consisted of data related to changes in protein thermostability induced by single mutations was retrieved from Protherm [30]. Only Protherm data that fulfilled the following four criteria were included: (1) proteins with a single mutation, (2) inclusion of temperature and pH values, (3) no missing attributes at each mutation site, and (4) inclusion of free-energy differences (ΔΔG). The resulting dataset, denoted S2864, that was used for training contained information for 868 stabilizing mutations (positive mutations) and 1996 destabilizing mutations (negative mutations). To eliminate possible bias in the training set induced by different numbers of positive and negative mutations, we created three datasets with ratios of 1:1, 1:1.5, and 1:2 of positive to negative mutations, each of which included all positive mutations. For each dataset, the destabilizing mutations were randomly selected and then tested for predictive accuracy 10 times, and the results are shown in Appendix A. We found that the positive to negative ratio of 1:2 resulted in the best Matthews correlation coefficient (*MCC*); therefore, the final training set, S2605, contained data for 868 positive and 1737 negative mutations. The resulting model then underwent a 10-fold cross-validation using the training set. An independent test set, denoted S438, was assembled by randomly selecting data for 146 positive and 292 negative mutations from test datasets in I-Mutant 2.0 [4] and PoPMuSiC [6]. Before randomly selecting these mutations, the data within I-Mutant 2.0 and PoPMuSiC were compared with the data in S2605 to eliminate redundant or conflicting data. The temperature values in S438 ranged from 5 to 65 °C and the pH values from 2.7 to 9.2.

### 2.2. Features

We included two types of features: basis and mRMR. Basis features are those that could be processed without feature selection (see below for details). mRMR features are the physicochemical attributes of amino acids, e.g., hydrophobicity and side-chain size, that would be selected by mRMR before processing to reduce computational load.

Basis features are defined as the wet-chemistry experimental characteristics used to determine stability, i.e., a wild-type residue and its replacement, location of the mutated residue, temperature, and experimental pH values. For the mRMR features, 544 physicochemical attributes of amino acids were retrieved from the AAindex database [31]. Because similar values included in the database may increase computational complexity, we calculated Pearson correlation coefficients between each attribute. The attributes with a correlation coefficient < 0.8 were retained. After filtering the attributes in this manner, their number was reduced to 371. Three types of features were generated from these attributes: (1) a total of 742 2-mRMR-type features, containing encoded amino-acid attributes for the wild-type and mutant proteins; (2) a total of 371 diff-mRMR-types features, containing numerical differences between amino-acid attributes of the wild-type and mutant proteins; and (3) a total of 1113 all-mRMR-type features, containing the 2-mRMR-type and diff-mRMR-type features.

### 2.3. mRMR Feature Selection

Redundancy and similar relationships between features will diminish the power of a tool meant to predict the effect a mutation will have on stability; in addition, a large number of features will increase computation time and computational complexity. Therefore, which features are selected must be carefully considered when training a prediction tool.

To date, the most widely used methods for feature selection have been InfoGain, which selects features that have the greatest weight at nodes on a decision-type tree, and Chi-Square, which uses a probability density function to select features. Peng and colleagues introduced the mRMR method for feature selection, which is more accurate than previous methods [24]. For mRMR feature selection, the method first compares the relevancy pairwise between features, corrects the results, and then compares the similarity between features. Relevant features that have corrected results are divided by their similarity to other features, with this value being the mRMR score that is used during training. This calculation ensures that the prediction method has the maximum relevance for the corrected results and minimum interference between features. The mRMR score is calculated by combining Equations (1) and (2) to produce Equation (3).
(1)$$\mathrm{Max}\;D=\frac{1}{\left|S\right|}\sum_{i\in S}I(i;c)$$
(2)$$\mathrm{Min}\;R=\frac{1}{{\left|S\right|}^{2}}\sum_{i,j\in S}I(i;j)$$
where *S* represents the feature set, *c* represents the target feature, *I*(*i*;*c*) represents the relevance between feature *i* and target feature *c*, and *I*(*i*;*j*) represents the relevance between features *i* and *j*.
(3)mRMR=D/R

All features are ranked according to their mRMR scores. Features with greater mRMR scores have a greater priority during construction of the model.

### 2.4. Regular-mRMR Feature Selection

Because a positive mutation to negative mutation ratio of 1:2 was used, some of the negative data in dataset S2864 were not included in training dataset S2605. Therefore, whether the data in S2605 were fully representative of the data in S2864 was uncertain. To address this issue, we developed the regular-mRMR method, which includes the following procedures.
Create *N* feature sets denoted as *PT_1~N_* by randomly selecting 868 samples *N* times from all data in S2864 (868 positive mutations and 1996 negative mutations).Perform mRMR-feature selection on each *PT* feature set.For each feature *j*, calculate the weight *k_i,j_* according to its mRMR score rank of each *PT_j_*, with *n* being the number of features to be chosen (Equation (4) below).The significance score *S_j_* is defined by the average *k_i,j_* (Equation (5) below). The priority of each feature is determined by the significance score.

The voting method is an ensemble method that improves the accuracy of a prediction model, without needing a complex computation algorithm, by combining multiple prediction results [32]. The voting method can also be used to select a machine-learning model [33]. The weighted voting model of regular-mRMR is based on the voting method.
(4)$$k_{i,j}=(n-pos_{j}(PT_{i})+1)$$
(5)$$S_{j}=Avg\sum_{i=1}^{n}k_{i,j}$$

In Equation (4), *n* is the number of features used, which is an arbitrary number. *pos_j_*(*PT_i_*) represents the ranking of feature *j* in the *i*-th feature set *PT_i_*. We first selected five feature sets *PT_1–5_* (*N* = 5), and, for each feature set, we selected the 50 highest-score features from mRMR for regular-mRMR. For the high-scoring features, 65% have mRMR scores > 0.4; therefore, we set n to 30. For each feature *j*, a weight value *k* was determined by the rank of its mRMR score. When feature *j* had the first rank in the dataset, its k value was 30, and when feature *j* had the n-th rank in the dataset, its *k* value was 31 − *n*. For each type of 2-mRMR, diff-mRMR, and all-mRMR feature, five feature sets were generated, and *S_j_* was calculated for each set (Equation (5)). The priorities of their mRMR features were then adjusted according to the *S_j_* value generated from the regular-mRMR values.

### 2.5. Hill-Climbing Algorithm

Our prediction model was constructed using basis features and regular-mRMR features. To select an adequate number of features from the regular-mRMR datasets, a hill-climbing algorithm was used [34]. For this algorithm, only the nodes closest to the current node are included in the comparison, and the search proceeds only when a new solution is better than the current node. Despite the speed and simplicity of the hill-climbing algorithm, it may not select the best solution for the model because the search may stop at a local, but not global, maximum. To prevent this problem, we started the hill-climbing procedure using the results calculated with the basis features, which improved the ability of the algorithm to reach the global maximum, thereby improving the accuracy of the method. Weka was used for the selection of classifiers, and we tested 58 classifiers for their performance in the basis feature set. The testing results for the classifiers are illustrated in Appendix A.

During the training of the classifiers, the feature with the greatest current score from the regular-mRMR test was added into the training set, and if it improved the *MCC*, the feature was retained as a classifier; otherwise, the feature was eliminated. The procedure was repeated until all features had been tested. The hill-climbing algorithm is given as Appendix A.

### 2.6. Evaluation

To assess the predictive performance of each classifier, we used the equations given below. *TP*, *FP*, *FN*, and *TN* represent true positive, false positive, false negative, and true negative values, respectively. Sensitivity (*Sn*), also denoted the true positive rate, reflects the percentage of correct predictions of changes in protein thermostabilities induced by mutations. Specificity (*Sp*), also denoted the true negative rate, reflects the percentage of correct predictions for destabilizations. Accuracy (*ACC*) is used to assess the overall predictive power of the prediction tool. *MCC* values range from −1 to 1, with a value of 1 representing a completely correct prediction, a value of 0 representing a random prediction, and a value of −1 representing exactly the opposite prediction, e.g., prediction of a destabilizing mutation when the mutation is stabilizing.
$$Sn=\frac{TP}{TP+FN}$$
$$Sp=\frac{TN}{TN+FP}$$
$$ACC=\frac{TN+TP}{TN+FN+TP+FP}$$
$$MCC=\frac{\left(TP\times TN)-(FP\times FN\right)}{\sqrt{\left(TP+FN)\times (TN+FP)\times (TP+FP)\times (TN+FN\right)}}$$

## 3. Results and Discussion

### 3.1. Ranking Features in the PT Files by mRMR

The 2-mRMR-type features were denoted with an initial “V” and are shown in Appendix A. Among these features, V645 was ranked first in four of the five tests run by mRMR. The priority rankings of the diff-mRMR-type features, denoted with an initial “M”, are also shown in Appendix A. Among these features, M324 was ranked first in all five tests. Priority feature rankings achieved by regular-mRMR are shown in Appendix A. M324, the most important feature in the diff-mRMR rankings, also ranked first among the all-mRMR features. However, other top-ranking all-mRMR-type features were ordered differently than in the other feature lists. For example, the feature V651, which ranked second in the all-mRMR list, ranked fifth in the 2-mRMR list. The feature V645, which ranked first in the 2-mRMR list, was not present in the all-mRMR list as it had a rank number > 30. Because there were marked differences in the rankings of the features in the all-mRMR lists and the other two lists, we needed to determine which feature list to use. Because the final model needed to incorporate all possible useful features, only those in the all-mRMR list were used in the construction of the final model.

### 3.2. Classifier Selection

The initial models were constructed with 58 different classifiers and trained with the basis features in the S2605 training set. The prediction qualities were evaluated via a 10-fold cross-validation. The results are shown in Appendix A. Seven types of classifiers were tested: bayes, function, lazy, meta, misc, rules, and trees. The classifiers that performed the best achieved an *ACC* < 0.8 and an *MCC* < 0.5 and were meta-type classifiers with 12 having an *ACC* > 0.7 and 7 having an *MCC* > 0.4.

The six best-performing algorithms are listed in Table 1 with their classifier. For the functional-type classifier, the algorithm that performed the best was LIBSVM [35], which achieved an *ACC* of 0.78 and an *MCC* of 0.43. For the tree-type classifier, the algorithm that performed the best was RandomForest [36], which achieved an *ACC* of 0.78 and an *MCC* of 0.47. The second best-performing tree-type algorithm was RandomTree [37], which achieved an *ACC* of 0.77 and an *MCC* of 0.45. For the meta-type classifier, the algorithm that performed the best was RotationForest [38], which achieved an *ACC* of 0.8 and an *MCC* of 0.51. For the lazy-type classifier, the algorithm that performed the best was KStar, which achieved an *ACC* of 0.81 and an *MCC* of 0.54. We chose KStar as our classifier for use in KStable because it had the best performance overall.

### 3.3. Basis Features Found Based on the Hill-Climbing Algorithm

The prediction models constructed using the hill-climbing algorithm with the regular-mRMR features are listed in Table 2. The basis model was constructed using only the basis features and had an *ACC* of 0.818 and an *MCC* of 0.562. After the addition of the first ranked feature from the regular-mRMR list of ranked features, M324, the *ACC* and *MCC* values decreased to 0.810 and 0.562, respectively. The poorer performance may have been caused by incompatibility between data selected from and not selected from the mRMR list. The incompatibility problem was partially solved by the addition of the mRMR feature V651. After the addition of V651, the *ACC* increased to 0.822, the *MCC* increased to 0.590, and the *Sn* value increased from 0.656 to 0.677. However, the *Sp* value was still less than that found using the model trained with only the basis features (0.894 compared with 0.906). The addition of the features V453 and M149 resulted in an improved *Sn* (0.708), *ACC* (0.827), and *MCC* (0.607). The addition of other features to a hill-climbing run did not exceed an *ACC* of 0.827 and an *MCC* of 0.607; therefore, KStable incorporated the features Basis, V651, V453, and M149 for construction.

### 3.4. Comparison of mRMR with Other Feature-Selection Methods

To compare the effects of regular-mRMR with other feature-selection methods, the top-30 ranked features were selected by the Infogain- and ChiSquared-Attribute Evaluation algorithms, under the same conditions used for the development of KStable. The selected features were then tested using hill-climbing, and the results revealed that the use of regular-mRMR achieved greater *ACC* and *MCC* values compared with the other two feature-selection methods (the results are given in Table 3).

### 3.5. Comparison of KStable with Other Predictors

Test set S438 was used to compare the performance of KStable with those of 13 other predictors (Table 4). The predictors EASE-MM, I-Mutant2.0_seq, INPS_seq, iPTREE-STAB, and MUpro require only a sequence as input. Only I-Mutant 2.0_seq, iPTREE-STAB, and KStable use temperature and pH data. The accuracy of pH–temperature ranges is shown in Appendix A, and the results of Appendix A indicate that KStable has a more stable prediction ability than the other sequence-based tools. AUTO-MUTE SVM, AUTO-MUTE RF, CUPSAT, DUET, I-Mutant2.0, MAESTRO, mCSM, PoPMuSiC, SDM, and SDM2 require structural data from the Protein Data Bank. KStable achieved an *ACC* of 0.708, an *MCC* of 0.298, an *Sn* of 0.411, and an *Sp* of 0.856 in S438. Compared with the other five sequence-based tools, the *Sn*, *ACC*, and *MCC* values of KStable were smaller than only those of EASE-MM, which achieved *Sn*, *ACC*, and *MCC* values of 0.658, 0.724, and 0.402, respectively. Despite being the best-performing sequence-based method, EASE-MM had the smallest *Sp* (0.757), and, owing to inclusion of position-specific scoring matrix-based evolution information, secondary structure prediction data from SPIDER, and disordered-region prediction data from SPINE-D, EASE-MM required more computational time than the other models. Moreover, among the six sequence-based methods, only KStable, I-Mutant 2.0_seq, and iPTREE-STAB included temperature and pH data as input, and KStable performed better than I-Mutant 2.0_seq. iPTREE-STAB achieved an *MCC* of 0.271—nearly the same as KStable—but had a worse *Sn* value (0.178). Compared with methods that require structural data, the *Sp* value calculated by KStable was smaller than only those of SDM (0.733) and SDM2 (0.616).

Because EASE-MM takes secondary structure and accessible surface area into account, to thoroughly compare the performances of EASE-MM and KStable, we examined the prediction results of both methods as a function of the secondary structural and accessible surface area information obtained from AUTO-MUTE. The results are shown in Table 5. We found that, when a mutation site is part of a coil or buried, KStable had better *MCC* and *Sp* values than EASE-MM. In addition, EASE-MM performed better for mutation sites located in strands than in helices, which is consistent with previous data.

### 3.6. Feature Analysis

The mRMR features V651 (β-sheet propensities derived from designed sequences), V453 (relative preference value at N), and M149 (transfer energy from an organic solvent/water) proved to be beneficial for the predictive power of KStable, when included in the hill-climbing model. Of all 30 potential mRMR features selected using the all-mRMR-type features, only four, V651, V453, V117, and V157, are features selected for the wild-type amino acid at the potential mutation site. The remaining 26 features are selected for the differences between the mutant and wild-type amino acids; that is, they are the diff-mRMR-type features. Notably, none of the included features are directly related to the mutant amino acid. Among the three characteristics included in the final KStable model, two are features of the wild-type amino acid, and only one involves a difference between the mutant and wild-type amino acids, which showed that, although the inclusion of diff-mRMR-type features in the regular-mRMR test yielded a better score than when 2-mRMR features were used, using diff-mRMR-type features did not necessarily improve the final model.

## 4. Conclusions

We developed KStable to predict changes in the thermostability of a protein upon mutation of a single residue in its sequence. KStable was trained using the following data: ΔΔG, temperature, and pH values; identities of the mutation site, and wild-type and mutant residues; and 544 physicochemical attributes of amino acids. The basis features, ΔΔG, temperature, pH, wild-type and mutant residues, and mutation site were processed without feature selection for training. The physicochemical attributes of the amino acids (the mRMR features) were first selected by the regular-mRMR method, which is based on the mRMR method and includes a procedure to eliminate possible conflicting characteristics in the training set and find representative features for the native data space. Compared with Infogain and ChiSquared, regular-mRMR achieved better *ACC* and *MCC* values for the feature selections.

Compared with other prediction models that require only sequence data as input, the *MCC* value of KStable was worse for only that of EASE-MM, whereas it was better than those of four other models. However, because EASE-MM includes a position-specific scoring matrix based on evolution information, secondary-structure prediction data from SPIDER, and disordered-region prediction data from SPINE-D, its computational feature-encoding time was greater than that for KStable. Moreover, EASE-MM does not use input based on pH and temperature conditions. However, protein thermostability changes are important for drug design and industrial applications [39,40]. KStable aims to rapidly predict the thermal stability of a target protein in various environments. Therefore, the purpose of EASE-MM is different from that of KStable; thus, the functionality of EASE-MM cannot be directly compared with that of KStable. Building a high-performance method for the prediction of protein thermal stability changes using sequence-based data is more difficult than that using structure-based data. One possible reason for this is that a prediction based on protein structure considers long-range interactions of the polypeptide chain [41]. However, compared with other prediction models that require structural data, KStable achieved a larger *MCC* value than four structural-data-requiring models and a better *Sn* value than eight structural-based models (only SDM and SDM2 yielded better values). We find KStable to be an accurate and fast computational tool that includes environmental conditions as input and can predict the thermostabilities of target proteins that have mutated at single sites.

## Figures and Tables

**Figure 1 entropy-20-00988-f001:**
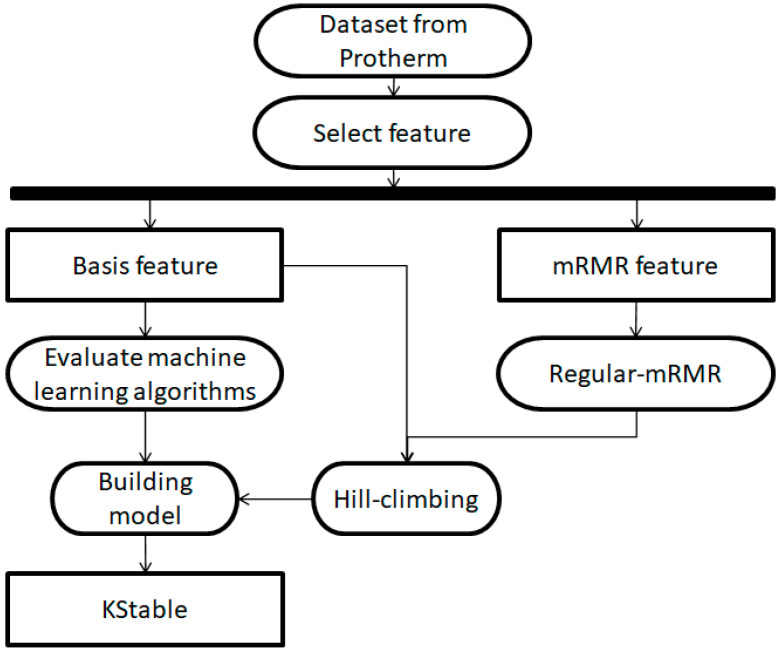
Workflow schematic for construction of KStable. mRMR, minimal redundancy–maximal relevance.

**Table 1 entropy-20-00988-t001:** The performance of the best classifiers in S2605.

Category	Classifier	*ACC*	*MCC*
function	LIBSVM	0.781	0.436
lazy	KStar	0.817	0.547
meta	RotationForest	0.805	0.513
rules	PART ^1^	0.788	0.478
trees	RandomForest	0.784	0.474
trees	RandomTree	0.774	0.459

^1^ PART: partial decision trees.

**Table 2 entropy-20-00988-t002:** Performance of the hill-climbing procedure when regular-mRMR (redundancy–maximal relevance) features were combined with the basis features.

Feature Combination	*Sn*	*Sp*	*ACC*	*MCC*
Basis	0.641	0.906	0.818	0.578
Basis + M324	0.656	0.887	0.810	0.562
Basis + V651	0.677	0.894	0.822	0.590
Basis + V651 + V453	0.687	0.888	0.821	0.590
Basis + V651 + V453 + M149	0.708	0.887	0.827	0.607

**Table 3 entropy-20-00988-t003:** Comparison of feature-selection methods.

Method	*Sn*	*Sp*	*ACC*	*MCC*
Regular-mRMR	0.708	0.887	0.827	0.607
InfoGain	0.683	0.896	0.825	0.597
ChiSquared	0.692	0.890	0.824	0.598

**Table 4 entropy-20-00988-t004:** Comparison of prediction results with the use of S438.

Method	*Sn*	*Sp*	*ACC*	*MCC*
**Sequence-based**				
KStable	0.411	0.856	0.708	0.298
EASE-MM	0.658	0.757	0.724	0.402
I-Mutant2.0_seq	0.185	0.918	0.674	0.151
INPS_seq	0.260	0.901	0.687	0.211
iPTREE-STAB	0.233	0.949	0.710	0.271
MUpro	0.267	0.901	0.689	0.218
**Structure-based**				
AUTO-MUTE SVM	0.075	0.969	0.671	0.101
AUTO-MUTE RF	0.137	0.976	0.696	0.222
CUPSAT	0.342	0.747	0.612	0.093
DUET	0.308	0.962	0.744	0.382
I-Mutant2.0	0.233	0.918	0.689	0.210
MAESTRO	0.342	0.921	0.728	0.334
mCSM	0.212	0.979	0.724	0.325
PoPMuSiC	0.247	0.955	0.719	0.302
SDM	0.733	0.736	0.735	0.448
SDM2	0.616	0.774	0.721	0.384

**Table 5 entropy-20-00988-t005:** Comparison of performances by KStable and EASE-MM, as correlated with secondary structures and solvent accessibility.

Structure *	Method	*Sn*	*Sp*	*ACC*	*MCC*
Coil	KStable	0.444	0.818	0.636	0.284
	EASE-MM	0.571	0.682	0.628	0.255
Helix	KStable	0.400	0.806	0.661	0.222
	EASE-MM	0.750	0.676	0.702	0.409
Strand	KStable	0.348	0.924	0.830	0.308
	EASE-MM	0.652	0.873	0.837	0.474
Buried	KStable	0.419	0.903	0.777	0.368
	EASE-MM	0.500	0.835	0.748	0.339
Surface	KStable	0.315	0.725	0.514	0.044
	EASE-MM	0.796	0.549	0.676	0.357
Under surface	KStable	0.567	0.831	0.747	0.405
	EASE-MM	0.733	0.708	0.716	0.414

* Structures defined by AUTO-MUTE.

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
