# Peer review of "KStable: A Computational Method for Predicting Protein Thermal Stability Changes by K-Star with Regular-mRMR Feature Selection"

_entropy, 2018, doi:10.3390/e20120988_

Reviewer 1 Report

The article "KStable: A Computational Method for Predicting Protein Thermal
Stability Changes by K-star With Regular-mRMR Feature Selection" submitted by Chi-Wei Chen et al, present an algorithm to predict the stability of proteins (wild-type and mutants) from their primary structure (sequence). There is a novelty on the way that the authors developed their algorithm, however when the author compared the performance of its algorithm they found that another software performs better in almost every calculation that they made. In order to improve their software, the author includes new features like pH and temperature. Even these features are desirable in the point of view of biology and also technology the author didn't show any graph or table where the robustness and dependence of their software with the pH or temperature. They must show how the software performs on know proteins when the pH and the temperature changes, otherwise there isn't any interest to use this software is other (better) exist and they are freely accessible on the web.

Author Response

Thanks for your constructive criticisms. As reviewer’s comment, comparative results with independent data from pH-temperature ranges were shown in table S7 in supplementary information, and the results of table S7 indicate KStable has more stable prediction ability than other tools.

Finally, some relative statements “The accuracy of pH-temperature ranges is shown in Table S7 and the results of Table S7 indicate KStable has more stable prediction ability than other sequence based tools” were added on line 278-279 in the manuscript.

Reviewer 2 Report

Dear Authors,

In the proposed methods only sequence information is used, instead of 3D data. However the influence of a certain amino acid replacement might depend on long range interactions of the polypeptide chain too. See for example: Magyar et al. BBRC 471 (1) 57-62. Kindly discuss how this issue influence your results and show a few results of your method and some experimental data.

Author Response

In the proposed methods only sequence information is used, instead of 3D data. However the influence of a certain amino acid replacement might depend on long range interactions of the polypeptide chain too. See for example: Magyar et al. BBRC 471 (1) 57-62. Kindly discuss how this issue influence your results and show a few results of your method and some experimental data.

Response:

Thanks for your constructive criticisms. Of course the protein 3D structure contains more information of amino acid interaction, and we also considered to add the sequence mimic 3D structure, but we finally do not use that information because these three points:

1.      Until now, the data of protein 3D structure are not abundant yet, but the mimic information are not fully correct with experimental data.

2.      KStable aims to rapidly predict thermal-stability of a target protein in various environments. If we applied long range interactions of the polypeptide chain by protein folding prediction, it could increase the period of system execution.

3.      We also compared with other prediction models that constructed with structural data. KStable achieved a higher MCC value than other four structure required models and a better Sn performance than eight structural-based models (only SDM and SDM2 yielded better values).

Finally, some relative statements were added on line 337-344 in the manuscript. “Building the high performance of protein thermal stability changes prediction by sequence based is more difficult than structure based. The one possible reason is prediction of base on protein structure considered long range interactions of the polypeptide chain [41].” However, compared with other prediction models that require structural data, KStable achieved a larger MCC value than found for four structure-requiring models and a better Sn value than eight structural-based models (only SDM and SDM2 yielded better values).

Reviewer 3 Report

Chen and co-authors present a predictor for Thermal stability called KStable. The compare their predictor to several other predictors and perform relatively well.

Major points:

EASE-MM is performing better; the authors do acknowledge this and justify the publication of KStable that it is more easy to use. Further, the authors claim that EASE-MM does not use pH and temperature as input. However, this is not sufficient. First, explain in greater detail why not using pH and temperature is an issue and provide the reader with a good sense of why this is important. Second, the authors are making it sound like using PSSMs, SPIDER and SPINE-D is somehow an issue. Why? These tools also only use sequence as input. Please do explain why this is an issue in detail.

Minor points:

At row 127, the authors write:

to be identical, and one of the two was
127 eliminated or merged with other identical attributes

How was one selected over the other? How were features merged? Why is this step even needed? Couldn't mRMR perform this automatically?

Author Response

Major points:

EASE-MM is performing better; the authors do acknowledge this and justify the publication of KStable that it is more easy to use. Further, the authors claim that EASE-MM does not use pH and temperature as input. However, this is not sufficient. First, explain in greater detail why not using pH and temperature is an issue and provide the reader with a good sense of why this is important. Second, the authors are making it sound like using PSSMs, SPIDER and SPINE-D is somehow an issue. Why? These tools also only use sequence as input. Please do explain why this is an issue in detail.

Response:

Thanks for your constructive criticisms. The responses are described as below:

1)      In this study, our purpose is rapidly predicting thermal-stability of a target protein in various environments, which could help bio-researcher in protein drugs design or industrial enzyme, and so on. Therefore, the purpose of EASE-MM and KStable is different.

Finally, for clarifying this issue, some relative statements were added as follow.

Line 278-279: “The accuracy of pH-temperature ranges is shown in Table S7 and the results of Table S7 indicate KStable has more stable prediction ability than other sequence based tools.”

Line 334-338: Moreover, EASE-MM does not use input based on pH and temperature conditions. “However, protein thermostability changes important for drug design and industrially applications [39,40]. KStable aims to rapidly predict thermal-stability of a target protein in various environments. Therefore, the purpose of EASE-MM and KStable is different, the functionality of EASE-MM can not be directly compared with that of KStable.”

2)      To use PSSM as encoding, we need to compare sequence to the NCBI Non-redundant protein sequence database (nr) by PSI-BLAST. This step cost much executed time. However, there are other ways could speed up the executed time. For example, the protein sequence alignment could only compare with human, one of species or training dataset, but current methods have limitation.

The SPIDER2 and SPIDER3 both use deep learning to predict secondary structure and accessible surface area, which have high predictive accuracy, but the SPIDER also applied PSSM as a feature-encoding, it would slowdown the totally executed time. It is necessary to wait for 21 minutes in the case of 107 amino acids as input (the version with PSSM http://sparks-lab.org/info/SPIDER3/2563/ and the version without PSSM http://sparks-lab.org/info/SPIDER3-Single/2067/).

SPOT-disorder (SPINE-D updated server) uses deep bidirectional long short-term memory recurrent neural networks to predict protein disorder, but the computation time is also very long.

3)      On the other hand, in the feature-encoding, for example, the sequence of amino acid is mutated from K to R, the mutation sequence will be a new query sequence to calculate by PSSM and structural prediction tool. And one site could change to other 19 types of amino acid, it requested that must have 19 calculations. It will take more time to make a full sequence prediction, let alone KStable contained pH and temperature parameters.

Minor points:

At row 127, the authors write:

to be identical, and one of the two was eliminated or merged with other identical attributes

How was one selected over the other? How were features merged? Why is this step even needed? Couldn't mRMR perform this automatically?

Response:

Thanks for your constructive criticisms.

We find that the original intention was not clear to described after English editing, and rewrite it.

We used Pearson correlation coefficients to filtered attributes could reduce computational complexity for further data processing, which include feature-encoding, mRMR, etc. Since, the model features of KStable have total of 1632 originally, and filtering will reduce 519 features.

Therefore, we rewrote the related description on line 124-126 in the manuscript “Because similar values included in the database may increase computational complexity, we calculated Pearson correlation coefficients between each attributes. The attributes with a correlation coefficient < 0.8 were retained”.

Round  2

Reviewer 1 Report

The authors of the manuscript titled "KStable: A Computational Method for Predicting Protein Thermal Stability Changes by K-star With Regular-mRMR Feature Selection" accomplismented the recommendent suggentions and I think that the manuscript can be accepted in the present form.